# A Cadaveric Study of Ulnar Nerve Movement and Strain around the Elbow Joint

**Mitsuyuki Nagashima** [1], **Shohei Omokawa** [2,*], **Yasuaki Nakanishi** [1], **Pasuk Mahakkanukrauh** [3,4], **Hideo Hasegawa** [1] and **Yasuhito Tanaka** [1]

1. Department of Orthopedic Surgery, Nara Medical University, 840 Shijo-cho, Kashihara, Nara 634-8521, Japan; m.nagashima0812@naramed-u.ac.jp (M.N.); savarez.alliance@gmail.com (Y.N.); hasehase08@icloud.com (H.H.); yatanaka@naramed-u.ac.jp (Y.T.)
2. Department of Hand Surgery, Nara Medical University, 840 Shijo-cho, Kashihara, Nara 634-8521, Japan
3. Department of Anatomy, Faculty of Medicine, Chiang Mai University, Chiang Mai 50200, Thailand; pasuk034@gmail.com
4. Excellence in Osteology Research and Training Center (ORTC), Chiang Mai University, Chiang Mai 50200, Thailand
* Correspondence: omokawa@gaia.eonet.ne.jp

**Abstract:** There is a lack of data on how ulnar nerve strain varies according to the location around the elbow joint. Therefore, we measured the longitudinal movement of the ulnar nerve around the elbow joint. Four fresh-frozen cadaveric upper extremities were used. A linear displacement sensor was attached to the ulnar nerve at eight measurement points at 20-mm intervals. At each point, the longitudinal movement of the ulnar nerve was measured during elbow flexion. We calculated the strain on the ulnar nerve based on the change in movement between neighboring points. Ulnar nerve movement with elbow flexion had a maximum value (mean, 10.5 mm; $p < 0.001$) at 2 cm proximal to the medial epicondyle. In the site distal to the medial epicondyle, the movement was small and demonstrated no significant difference between points ($p = 0.1$). The change in strain between mild flexion ($0$–$60°$) and deep flexion ($60$–$120°$) significantly differed at 2–4 cm and 6–8 cm proximal to the medial epicondyle (15% versus 3%, $p < 0.01$; 5% versus 9%, $p < 0.05$, respectively). The longitudinal movement of the ulnar nerve during elbow flexion occurred mainly at the site proximal to the medial epicondyle and became smaller away from the medial epicondyle.

**Keywords:** cadaveric study; elbow flexion; elbow joint; movement; strain; ulnar nerve

## 1. Introduction

Peripheral nerves need to glide and stretch to adjust to the changes in nerve travel associated with joint movement [1]. Fifteen percent elongation of a peripheral nerve reduces the nerve blood flow by 80% [2]. In addition, prolonged strain on the nerves causes irreversible nerve conduction disorders [3]. Flexion of the elbow joint increases the distance traveled by the ulnar nerve [1]. The ulnar nerve stretches during elbow flexion, and the overstrain of the nerve may be involved in the development of cubital tunnel syndrome [4–6]. Furthermore, amateur and professional baseball players frequently experience cubital tunnel syndrome [7]. There have been many studies on ulnar nerve sliding and strain using cadavers [1,5–12]. Vinitpairot et al. measured ulnar nerve strain at the posterior medial epicondyle of the humerus during different daily activities and reported that daily activities did not cause permanent damage to the nerve [12]. However, ulnar neuropathy has similarly been reported to occur in the arcade of Struthers, which is far from the cubital canal [13]; nonetheless, there is a lack of data on how ulnar nerve strain varies according to the location around the elbow joint. The establishment of normal ulnar nerve movement is important for it to be compared with movements in painful conditions, such as cubital tunnel syndrome and traumatic nerve injury, that affect the ulnar nerve.

We hypothesized that nerve strain would increase at a location closer to the elbow joint and that nerve movement would be larger in the proximal site of the elbow joint than in the distal site. In this study, we measured the longitudinal movement of the ulnar nerve around the elbow joint during elbow motion using fresh-frozen cadavers.

## 2. Materials and Methods

### 2.1. Ethics Considerations

The present study on human cadavers was approved by the Research Ethics Committee, Faculty of Medicine, Chiang Mai University (Chiang Mai, Thailand, code: ANA-2561-06030). The Department of Anatomy at Chiang Mai University provided the cadavers for this study. The living patients provided informed consent for their cadavers to be used for research purposes.

### 2.2. Specimen Preparation

We used four fresh-frozen cadaveric upper extremities from two men and two women with an average age of 70 years (range, 50–93 years) at the time of death. None of the specimens had any trauma or deformity of the neck, shoulder, or upper extremity. After fixing the ulnar nerve to the humeral head in a fixed limb position, the shoulder joint was in 60° abduction and a neutral rotation position, the elbow joint was in full extension, and all specimens were amputated at the shoulder joint. Skin and subcutaneous tissue were removed from the upper arm to the middle forearm; the tissue deep to the fascia was preserved. The forearm and wrist joint were fixed in the neutral position with Kirschner wires inserted into the distal radioulnar joint and the radiocarpal joint, respectively. Each upper limb specimen was placed on the experimental table with an external fixator (Figure 1).

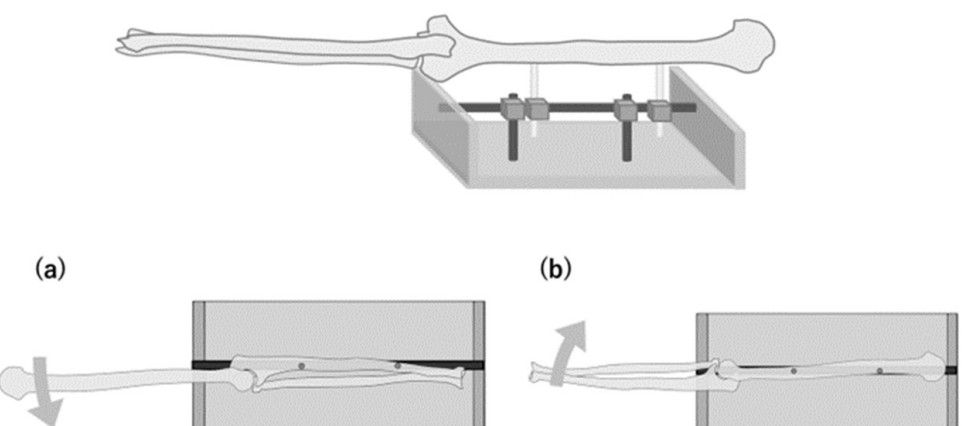

**Figure 1.** Schematic illustration of specimen fixing. The ulna is fixed to the experimental table when measuring the movement of the ulnar nerve proximal to the medial epicondyle (**a**). The humerus is fixed to the experimental table when measuring the movement distal to the medial epicondyle (**b**).

### 2.3. Measurement of Ulnar Nerve Movement

With the elbow at full extension, eight measurement points (p20, p40, p60, p80, d20, d40, d60, and d80) were set at 20-mm intervals, which were located proximal and distal to the medial epicondyle and centered around the medial epicondyle (Figure 2). A linear displacement sensor (Pulse-Coder; LEVEX, Kyoto, Japan), which comprised a coil sensor and a brass pipe, was attached to the ulnar nerve at each point. The measurement system has been described previously [1], and the range of measurement was 14 mm. The coil sensor was fixed to the external fixator. The 20-mm-long needle that was attached to the pipe was inserted into each point of the ulnar nerve. The needle had a barb to ensure that it did not slip off the nerve (Figure 3). The needle was placed on the nerve through a small window in the fascia, and care was taken to preserve the surrounding tissue. The change

in the linear displacement of the pipe when the elbow was flexed was measured as the change in the longitudinal movement of the nerve.

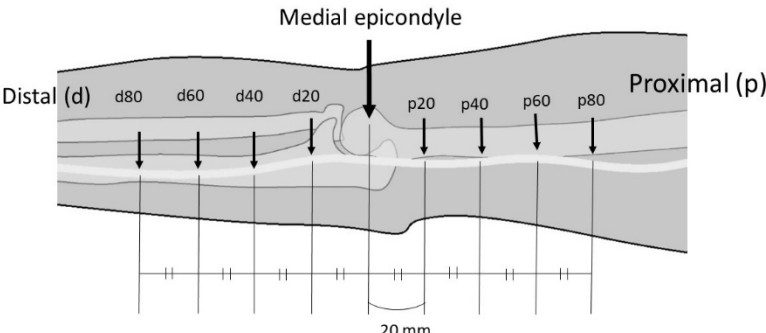

**Figure 2.** Measurement points on the ulnar nerve. Eight points (p20, p40, p60, p80, d20, d40, d60, and d80) set at 20-mm intervals are located proximal (p) and distal (d) to the medial epicondyle and at the center of the medial epicondyle in the elbow extension position.

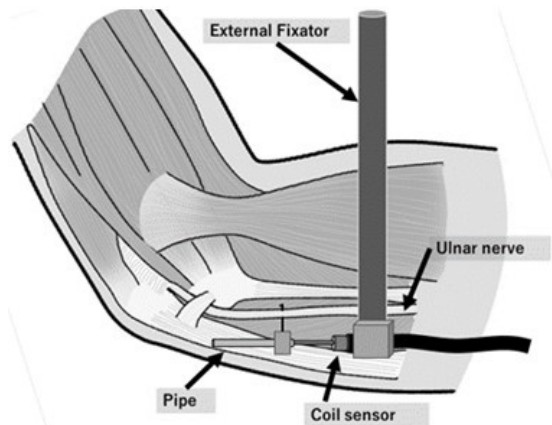

**Figure 3.** Schematic illustration of the linear displacement sensor mounted at each point of the ulnar nerve. A barb on the needle prevents the pipe from slipping off the nerve. The coil sensor is fixed by an external fixator.

*2.4. Experimental Sequences*

Two φ5.0-mm stainless pins were inserted into the humerus and fixed to the experimental table with an external fixator when measuring the changes in the longitudinal movement of the ulnar nerve proximal to the medial epicondyle. Meanwhile, when measuring them distal to the medial epicondyle, two stainless steel pins were inserted into the ulna and fixed to the table (Figure 1). At each point, changes in the longitudinal movement were measured during elbow flexion at 0°, 30°, 60°, 90°, 120°, and maximum flexion. The same measurement was performed three times, and the average value was used. Ulnar nerve strain was calculated from the absolute change in movement between neighboring points (six sections: p20–40, p40–60, p60–80, d20–40, d40–60, and d60–80).

For example, strain of p20–40

$$\text{strain} = \frac{\text{Mp20} - \text{Mp40 (mm)}}{20 \text{ mm (sectional distance)}} \times 100 \ (\%)$$

Mp20: change in movement at p20
Mp40: change in movement at p40

### 2.5. Statistical Analysis

A one-way repeated analysis of variance was performed on the movement and strain values, followed by the Tukey–Kramer test. The strain values were compared between mild flexion (0–60°) and deep flexion (60–120°) of elbow flexion angles using paired *t*-tests. SPSS Statistics for Windows (version 26.0; IBM Corp., Armonk, NY, USA) was used to conduct statistical analyses, and statistical significance was set at $p < 0.05$.

### 3. Results

During elbow flexion, the ulnar nerve moved peripherally at the site proximal to the medial epicondyle and centrally at the site distal to the medial epicondyle. Ulnar nerve movement with elbow flexion had a maximum value at the site proximal to the medial epicondyle, demonstrating an average of 10.5 mm at p20 (maximum flexion: p20, 10.5 mm; p40, 6.9 mm; p60, 5.4 mm; p80, 2.6 mm, $p < 0.001$; Figure 4a). At the distal site of the medial epicondyle, the movement was small and exhibited no significant difference between points (maximum flexion: d20, 2.3 mm; d40, 1.1 mm; d60, 1.0 mm; d80, 0.6 mm; $p = 0.10$; Figure 4b).

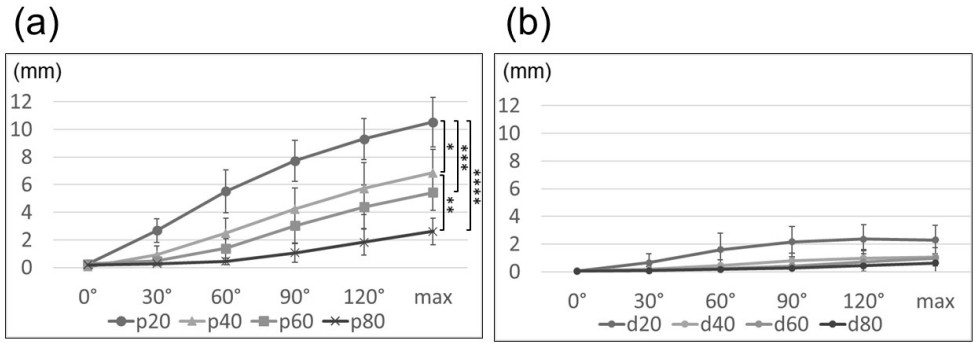

**Figure 4.** Line graph illustrating the movement of the ulnar nerve proximal to the medial epicondyle (**a**) and distal to the medial epicondyle (**b**). *: p20–p40, **: p40–p80, ***: p20–p60, ****: p20–p80 (*: $p < 0.05$, **: $p < 0.01$, ***: $p < 0.05$, ****: $p < 0.001$). max, maximum.

Ulnar nerve strain from extension to maximum flexion demonstrated no significant difference in each section, either proximal or distal to the medial epicondyle ($p = 0.30$ and $p = 0.08$, respectively; Figure 5). However, strain revealed the maximum value in the section of p20–40 and had an 18% increase in maximum flexion compared with elbow extension; additionally, the change in strain during mild flexion (0–60°, 15%) was significantly larger than that during deep flexion (60–120°, 3%; $p < 0.05$) (Figure 6). In contrast, strain in the section of p60–80 changed by 5% and 9% during mild and deep flexion, respectively ($p < 0.05$).

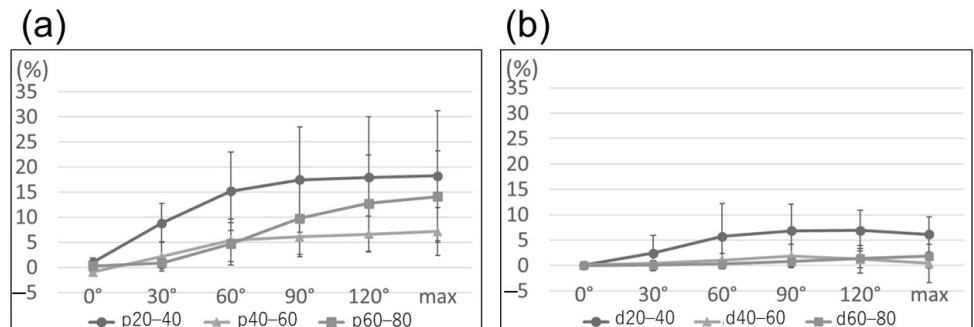

**Figure 5.** Line graph illustrating ulnar nerve strain proximal to the medial epicondyle (**a**) and distal to the medial epicondyle (**b**). There is no significant difference in each section at maximum flexion of the elbow proximal and distal to the medial epicondyle ($p = 0.30$ and $p = 0.08$, respectively).

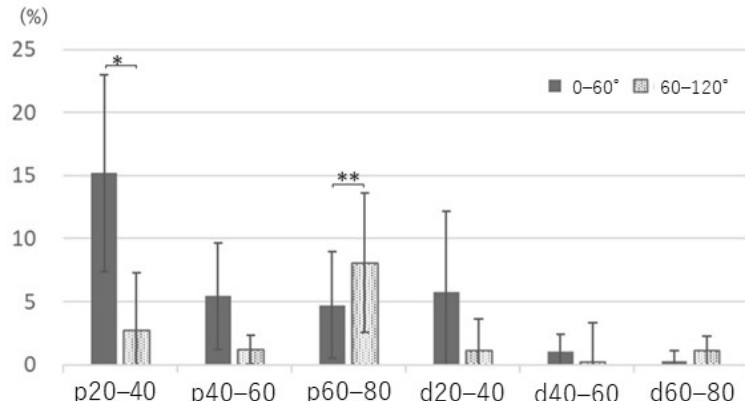

**Figure 6.** Comparison of ulnar nerve strain with elbow flexion divided into mild flexion (0–60°) and deep flexion (60–120°). p20–40: 15% versus 3%, *: $p < 0.01$; p60–80: 5% versus 9%, **: $p < 0.05$.

## 4. Discussion

The ulnar nerve slides during elbow motion and stretching, and nerve slacking repeatedly occurs [6,8]. Nerve strain of 5–10% impairs intraneural blood flow [2,14,15], axonal transport [16], and nerve conduction [3,17]. Recent studies wherein a strain gauge was used reported that ulnar nerve strain changes at different upper limb positions. Aoki et al. reported that the greatest ulnar nerve strain is observed during the acceleration phase of the pitching motion [1]. Wright et al. found that composite upper extremity joint movement caused more than 15% strain on the ulnar nerve [6]. Toby et al. observed that the maximum strain was behind the medial epicondyle during maximal elbow flexion [9]. Hicks et al. and Cirpar et al. measured strain changes after in situ decompression and medial epicondylectomy and showed that the medial epicondyle plays a major role in increasing ulnar nerve strain [10,11]. Although Vinitpairot et al. did not use strain gauges, they reported that daily activities involving the elbow and shoulder joints increased the strain on the ulnar nerve; however, no permanent nerve damage was observed [12]. Conversely, these studies measured only one site in or near the cubital tunnel. Mahan et al. measured nerve strain by placing markers on the ulnar nerve over a wide area of the upper arm and forearm [18]. They removed the fascia and Osborne ligament around the nerve to place the markers, which may be far from the in vivo environment. Thus, it remains unclear how the strain changes depending on the location around the elbow joint.

The present study measured the longitudinal movement of the ulnar nerve at several sites near the elbow by preserving the soft tissue around the ulnar nerve as much as possible. The results revealed that ulnar nerve movement differs according to the measuring site, and the degree of strain changes across sites at the elbow joint position. Proximal to the medial epicondyle, movement of the ulnar nerve during elbow flexion increased nearer to the elbow and decreased with an increase in the distance from the elbow. This result is consistent with our hypothesis. Aoki et al. observed that the ulnar nerve moves 12.4 mm with maximum elbow flexion at the site 3 cm proximal to the cubital tunnel [1]. Our study demonstrated that at 2 cm proximal to the medial epicondyle, the ulnar nerve moved 10.4 mm with full elbow flexion. These findings suggest that at least 10 mm of ulnar nerve movement occurs on the proximal side of the medial epicondyle with elbow flexion.

Ulnar nerve strain proximal to the medial epicondyle demonstrated the maximum value near the elbow (p20–40). Major strain associated with elbow flexion near the elbow (p20–40) occurred before 60° of elbow flexion; however, major strain in the more proximal section (p60–80) occurred with deep flexion of 60° or more. This result did not support our hypothesis; nevertheless, it was an interesting finding. Aoki et al. reported that major strain change in the ulnar nerve at the site 3 cm proximal to the cubital tunnel occurred before 90° of elbow flexion, which is consistent with the results of this study [1].

The ulnar nerve distal to the medial epicondyle moved proximally with elbow flexion; however, the movement was small compared with the proximal part of the medial

epicondyle. Dilley et al. used an ultrasound machine to measure sliding of the ulnar nerve in elbow flexion and reported slight sliding of the nerve in the forearm [19,20]. Similar results were obtained in our study. Paulos et al. reported that the ulnar nerve distal to the elbow has two to five muscular branches to the flexor carpi ulnaris and one to two muscular branches to the flexor digitorum profundus [21]. The ulnar nerve may be strongly anchored to the forearm flexors by multiple branches. We assumed that these branches fixed the ulnar nerve distal to the medial epicondyle and suppressed the movement and strain of the nerve with elbow flexion.

Dilley et al. also reported little or no nerve sliding in the upper arm. In this study, nerve movement was small both at 8 cm proximal to the medial epicondyle and in the forearm. Dilley et al. attributed this phenomenon to the non-straight, snaking segment of the nerve that reduces the load on the nerve caused by the flexion movement of the elbow, which was referred to as the 'high compliance segment' [19]. Straightening the high compliance segment near the elbow reduces the effect of elbow flexion. Consequently, the effect on the segment away from the elbow may be reduced, as would the amount of movement away from the elbow.

The findings of this study indicate that ulnar neuropathy due to over-traction of the valgus elbow may occur on the proximal side of the elbow. When a nerve is inflamed, even a 3% increase in nerve strain can produce ectopic impulses, which can cause pain and paresthesia [22]. Our findings suggest that the inching electrodiagnosis method [23] may be useful for early neuropathy due to over-traction. This study was performed at 60° of shoulder abduction, and the upper limb position at maximum elbow flexion was similar to the upper limb position of the motion of eating with a spoon (70° of shoulder abduction and 130° of elbow flexion), which is a daily activity with maximum ulnar nerve strain, as reported by Vinitpairot et al. [13]. They reported that this motion caused 7% to 19% strain when the gliding of the ulnar nerve was restricted. In this study, we observed 18% ulnar nerve strain even in the presence of ulnar nerve gliding, which is close to the value they reported with restricted nerve gliding. They excised the fascia at the measurement site, whereas in our study, a small window was created only at the insertion site of the sensor needle to preserve the fascia, and the tension of the fascia associated with joint movement may have affected the increase in ulnar nerve strain. Wall et al. reported that a 12% strain on the nerve lasting for 1 h caused nerve conduction disturbance with only a minimal recovery [3], and prolonged continuation of a limb position that increases the strain on the ulnar nerve even in daily activities may cause nerve damage. In addition, restriction of ulnar nerve gliding increases nerve strain by 50 to 154% compared to the condition with nerve gliding [12,18]. Therefore, ulnar nerve adhesions near the elbow due to trauma or surgical manipulation are more likely to cause nerve over-traction than those at sites far from the elbow. In our study, the ulnar nerve exhibited considerable movement at 2–4 cm proximal to the medial epicondyle and 14% strain 6–8 cm proximal to the medial epicondyle in deep elbow flexion; therefore, it may be necessary to devise strategies for the prevention of nerve adhesions on a wide range, such as nerve wrapping [24,25]. In the future, revealing the changes in strain due to nerve adhesions around the elbow over a wide area of the ulnar nerve may aid in identifying areas where nerve adhesions need to be prevented. This may help prevent ulnar nerve neuropathy in the elbow, and further studies are thus required.

This study has several limitations. First, we observed differences in ulnar nerve strain at different sites; however, the sample size was small. Second, caution was exercised not to damage the soft tissues, such as para-neurium, that are crucial for nerve gliding [26] during the study; however, over time, the properties of the fascia and nerves may change due to drying, and the environment may become different from that in vivo. Third, we amputated all specimens at the shoulder joint and fixed the proximal ulnar nerve to the humeral head. We used a suture and a screw to fix the ulnar nerve to the humeral head, which may not recreate the normal physiological resting nerve tension. To measure nerve strain in an environment closer to the living body, it was necessary to leave it to

the cervical vertebrae and maintain continuity between the peripheral nerves and the spinal cord. Finally, although we could measure the precise amount of movement with our sensor, it could only measure the linear movement displacement, and we could not evaluate movement within 2 cm from the medial epicondyle because the nerve bent with elbow flexion.

## 5. Conclusions

The longitudinal movement of the ulnar nerve following elbow flexion occurred mainly proximal to the medial epicondyle. Ulnar nerve movement was maximal 20 mm proximal to the medial epicondyle and became smaller away from the medial epicondyle. This study revealed that ulnar nerve strain with elbow flexion and the elbow joint angle at which the strain changes differ depending on the region.

**Author Contributions:** Conceptualization, S.O.; methodology, M.N.; investigation, M.N., Y.N. and H.H.; writing—original draft preparation, M.N.; project administration, S.O.; validation, Y.N.; formal analysis, Y.N.; resources, P.M.; visualization, H.H.; supervision, Y.T. All authors have read and agreed to the published version of the manuscript.

**Funding:** This research received no external funding.

**Institutional Review Board Statement:** The study was conducted according to the guidelines of the Declaration of Helsinki and was approved by the Chiang Mai University Faculty of Medicine Research Ethics Committee (Chiang Mai, Thailand; code: ANA-2561-06030; date of approval: 7 January 2019).

**Informed Consent Statement:** Consent for the use of these cadavers was obtained from the patients before death.

**Data Availability Statement:** The data presented in this study are available in the article.

**Acknowledgments:** The authors thank the staff of the Excellence in Osteology Research and Training Center (ORTC), Surgical Training Center, and the Departments of Anatomy and Orthopedics of Chiang Mai University for their assistance in this study.

**Conflicts of Interest:** The authors declare no conflict of interest.

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
