# Peer review of "A Cadaveric Study of Ulnar Nerve Movement and Strain around the Elbow Joint"

_applsci, doi:10.3390/app11146487_

Round 1
Reviewer 1 Report
This cadaveric study using fresh-frozen cadaveric upper limbs assessed the longitudinal displacement of the ulnar nerve during the flexion of the elbow.
The introduction contains the background of the study, objectives and hypothesis; Material and methods explains the ethical considerations, the preparation of the arms, how they performed the measurements and statistical analyses; Results are well presented and Discussion showed their main findings, limitations and clinical implications.
Congratulations on the work you have done
Author Response
 Thank you very much for reviewing our manuscript.
Reviewer 2 Report
The article is well written and has a proper hypothesis which the authors have addressed by their work. They have clearly explained their findings and have listed the limitations of the study.
Lines 165-166 need clarification to make its meaning clear.
Author Response
Thank you very much for reviewing our manuscript.
In Lines 165-166, I mentioned that there was less sliding of the ulnar nerve in the forearm, citing Dilley et al.
I have changed the position of the text to make it easier to understand the cause of this result, which was discussed later in the paper, citing Paulos et al.
Text after the change
“The ulnar nerve distal to the medial epicondyle had moved proximally with elbow flexion. The movement was small compared with the proximal part of the medial epicondyle. Dilley et al. used an ultrasound machine to measure sliding of the ulnar nerve in elbow flexion and reported slight sliding of the nerve in the forearm [15,16]. Similar results were obtained in our study. Paulos et al. reported that the ulnar nerve distal to the elbow has two to five muscular branches to the flexor carpi ulnaris and one to two muscular branches to the flexor digitorum profundus [17]. The ulnar nerve may be strongly anchored to the forearm flexors by multiple branches. We assumed that these branches fixed the ulnar nerve distal to the medial epicondyle and suppressed the movement and strain of the nerve with elbow flexion.”
Reviewer 3 Report
This cadaver study reports movement and tension of the ulnar nerve around the elbow joint. Shoulder and elbow movements can affect ulnar nerve tension. However, there is no evidence linking this type of strain to specific activities. Elbow flexion and shoulder abduction in daily activities are associated with increases in ulnar nerve strain, but this may not cause permanent damage to the nerve. After nerve gliding motion had been restricted, nerves that normally exhibited less strain often had even increased higher levels of strain than those nerves that normally exhibited high strain (cit. Vinitpairot, 2019).
In several works it is described the tension of the ulnar nerve at the elbow due to related to common daily activities movement that was measured both in normal nerves that in the nerves in which the sliding movement was limited.
This paper refers only to the normal ulnar nerve.
The bibliography of the paper is not complete and is missing some important references that would be useful to verify the reliability of the results obtained compared with those also evaluated by other authors
The limitations described by the Authors are many and important and greatly affect this study including results.
Author Response
Thank you very much for reviewing our manuscript. Thank you also for introducing the reference.
Accordingly, we have cited some studies, including the one you suggested, and revised the Discussion based on these papers to show the reliability of this study. In addition, I have added a relevant description of a limitation regarding improvements in this study.
Reviewer 4 Report
In their study, the authors examine the gliding properties of the ulnar nerve at the elbow level in fresh frozen human cadavers. The methods are clearly depicted and the results precisely shown.
They find the main movement proximal to the medial epicondyle, whereas distal to it the nerve seems to move only little.
Open questions:
- what is the role of Osbourne ligament? Usually its considered to be the site of major compression and thus fixation of the nerve? Could this be a reason for the distal non-movement?
- discussion: the authors suggest a "nerve wrapping" at the high-movement level. I doubt there enough evidence for this hint in this study. A lot of artificial "nerve wraps" have been postulated. Being a foreign body, most of them lead to severe scarring and worsening of gliding ability.
- discussion: how should nerve adhesions be prevented?
- According to one of nerve surgery's pioneers, Prof. Millesi (Vienna, Austria), the so called para-neurium is a key factor for nerve gliding properties. This layer is on the surface of the epineurium allowing for gliding movements between the nerve and its surrounding tissue. Was this layer dissected out in the study? (Millesi H, Zöch G, Rath T. The gliding apparatus of peripheral nerve and its clinical significance. Ann Chir Main Memb Super. 1990;9(2):87-97. doi: 10.1016/s0753-9053(05)80485-5. PMID: 1695518)
Author Response
Thank you very much for reviewing our manuscript.
Answer in order.
Point 1: what is the role of Osbourne ligament? Usually its considered to be the site of major compression and thus fixation of the nerve? Could this be a reason for the distal non-movement?
Response 1: Elbow flexion causes the ulnar nerve to bend behind the medial epicondyle. If the ulnar nerve is fixed by the Osborne ligament, it is not expected to experience any strain distal to the Osborne ligament; however, in this study, we observed approximately 5% strain even 2 cm distal to the medial epicondyle. Vinitpairot et al. reported an increase in strain by anchoring the ulnar nerve to the Osborne ligament (reference 12). If the nerve is entrapped in the Osborne ligament, as you suggested, the nerve may be immobilized; however, under normal conditions, we believe that the Osborne ligament is a mobile supportive tissue that maintains the ulnar nerve in the cubital tunnel.
Point 2: discussion: the authors suggest a "nerve wrapping" at the high-movement level. I doubt there enough evidence for this hint in this study. A lot of artificial "nerve wraps" have been postulated. Being a foreign body, most of them lead to severe scarring and worsening of gliding ability.
Response 2: As you indicated, our study was not exclusively sufficient to demonstrate the necessity of nerve wrapping; hence, we added the studies by Vinitpairot et al. and Mahan et al. to show that nerve strain increases when nerve adhesion occurs (references 12 &13).
Point 3: discussion: how should nerve adhesions be prevented?
Response 3: Thank you for introducing a study to improve the quality of our manuscript. In our study, we minimally perforated the fascia to insert the needle into the nerve and did not dissect the para-neurium. Accordingly, we have cited a reference related to this description and added that it maintains structures important for nerve gliding in our manuscript.
Point 4: According to one of nerve surgery's pioneers, Prof. Millesi (Vienna, Austria), the so called para-neurium is a key factor for nerve gliding properties. This layer is on the surface of the epineurium allowing for gliding movements between the nerve and its surrounding tissue. Was this layer dissected out in the study? (Millesi H, Zöch G, Rath T. The gliding apparatus of peripheral nerve and its clinical significance. Ann Chir Main Memb Super. 1990;9(2):87-97. doi: 10.1016/s0753-9053(05)80485-5. PMID: 1695518)
Response 4: Thank you for introducing a study to improve the quality of our manuscript. In our study, we minimally perforated the fascia to insert the needle into the nerve and did not dissect the para-neurium. Accordingly, we have cited a reference related to this description and added that it maintains structures important for nerve gliding in our manuscript.

Round 2
Reviewer 3 Report
Thanks for the review. The additions allow us to better understand the experimental design. The updating of the bibliography and references adequately completes the work